# Environmental Barriers and Functional Outcomes in Patients with Schizophrenia in Taiwan: The Capacity-Performance Discrepancy

**DOI:** 10.3390/ijerph19010315

**Published:** 2021-12-28

**Authors:** Wei-Chih Lien, Wei-Ming Wang, Hui-Min David Wang, Feng-Huei Lin, Fen-Zhi Yao

**Affiliations:** 1Department of Physical Medicine and Rehabilitation, National Cheng Kung University Hospital, College of Medicine, National Cheng Kung University, Tainan 704, Taiwan; lwclwhab@ms8.hinet.net; 2Department of Physical Medicine and Rehabilitation, College of Medicine, National Cheng Kung University, Tainan 701, Taiwan; 3Ph.D. Program in Tissue Engineering and Regenerative Medicine, National Chung Hsing University, Taichung 402, Taiwan; davidw@dragon.nchu.edu.tw; 4Department of Statistics, College of Management, National Cheng Kung University, Tainan 701, Taiwan; weiming3524@gmail.com; 5Graduate Institute of Biomedical Engineering, National Chung Hsing University, Taichung City 402, Taiwan; 6Institute of Biomedical Engineering, College of Medicine and Engineering, National Taiwan University, Taipei 100, Taiwan; 7Institute of Biomedical Engineering and Nanomedicine, National Health Research Institutes, Zhunan, Miaoli 350, Taiwan; 8Department of Senior Citizen Services, National Tainan Junior College of Nursing, Tainan 700, Taiwan

**Keywords:** schizophrenia, environmental barriers, capacity, performance

## Abstract

Environmental factors are crucial determinants of disability in schizophrenic patients. Using data from the 2014–2018 Certification of Disability and Care Needs dataset, we identified 3882 adult patients (46.78% females; age, 51.01 ± 13.9 years) with schizophrenia. We found that patients with severe schizophrenia had lower capacity and performance than those with moderate schizophrenia. The chances of having an access barrier to environmental chapter 1 (e1) products and technology in moderate schizophrenic patients and in severe schizophrenic patients were 29.5% and 37.8%, respectively. Logistic regression analyses demonstrated that the performance score was related to accessibility barriers in the categories described in e1, with adequate fitness of models in category e110 for personal consumption, e115 for personal usage in daily living activities, and e120 for personal outdoor and indoor mobility and transportation. Furthermore, the capacity-performance discrepancy was higher in moderate schizophrenic patients with accessibility barriers in the e110, e115, and e120 categories than that in moderate schizophrenic patients without accessibility barriers. However, severe schizophrenic patients with category e120 accessibility barriers were prone to a lower discrepancy, with institutional care a potentially decreasing factor. In conclusion, providing an e1 barrier-free environment is necessary for patients with schizophrenia to decrease their disability.

## 1. Introduction

Chronic mental health conditions are one of the major causes of disability among disabled individuals in the United States [1]. This is also true in Taiwan, with a prevalence rate of 10–15% among the disabled population, followed by moving functional limitation and internal organ of loss function and related disabilities according to the Ministry of Health and Welfare (MOHW) of Taiwan, Republic of China (R.O.C.) [2]. Notably, schizophrenia is one of the major causes of disability among chronic mental health conditions, affecting approximately 1% of the world’s population [3] and around 50% of the population with psychiatric disabilities [4]. In fact, schizophrenia is one of the 20 most debilitating diseases, with a prevalence of 26.3 million patients worldwide [5]. It not only impairs cognitive abilities but also affects numerous crucial and major physical activities, including self-care, mobility function, and participation in society [4]. Furthermore, schizophrenia results in a huge financial burden on individuals, families, and society [6]. While psychosocial disability is not regarded as an unmediated outcome of schizophrenia, it is potentially mediated by environmental contextual factors [7]. However, a recent review reported that the majority of the studies to date have largely focused on body function and activities/participation chapters in agreement with the International Classification of Functioning, Disability and Health (ICF) model in schizophrenia instead of contextual factors, including personal and environmental factors [8].

The standardized functional assessment in patients with schizophrenia is clinically meaningful. The Global Assessment of Functioning (GAF) is a reliable and validated instrument for global psychotic and social functional impairment in patients with schizophrenia within the ICF model [9]. Based on the biopsychosocial concept of the ICF, the development of the second edition of the World Health Organization Disability Assessment Schedule (WHODAS 2.0) was aimed at determining the difficulty of daily living and participation in society confronted by a patient in the preceding 30 days. The WHODAS 2.0, which is a validated and useful measure of disability for schizophrenia [10], assesses functional disability through both capacity (what a patient can accomplish in a controlled, uniform situation) and performance (what a patient can actually accomplish in his/her everyday situation) [11]. The evaluation of both capacity and performance in patients with schizophrenia is clinically relevant [12]. The WHO highlighted the crucial impact of contextual factors, including environmental and personal factors, on a person’s functional outcomes. Taking environmental factors into account confirmed that disablement is no longer considered a feature of a person but as the relationship of the person with a disease and its surroundings. In a cohort study using data from the Medicare Current Beneficiary Survey [13], the living arrangement was identified as an important factor for functional deterioration in the elderly population. Environmental barriers may influence people’s difficulty in activities of daily living through decreasing their participation and making them less autonomous [14]. Notably, the capacity-performance discrepancy was found to be associated with personal factors, such as personality traits, in schizophrenia [15]. In a special article [16], the concept of environmental factors in schizophrenia was proposed. However, whether environmental factors (Exposure) affect the capacity-performance discrepancy (Outcome) in schizophrenia (Population) remains unknown. Table 1 shows the PECO (Population, Exposure, Comparator, and Outcome) statement of the study.

The environment can play a pivotal role in influencing the degree of social participation in stroke patients [17], and it is associated with the function of daily activities in older adults [18]. These relationships are considered to operate in both directions that are present between ICF chapters [19,20]. Furthermore, environmental factors that are involved in the physical, social, and attitudinal ambience of persons affected with schizophrenia also produce barriers and enablers [21]. Environmental factors can be divided into the following chapters: first chapter (environmental chapter 1 (e1), products and technology), second chapter (e2, natural environment), third chapter (e3, support and relationships), and fourth chapter (e4, attitudes), with e1 accessibility barriers being the most encountered and addressed in schizophrenia [8]. However, whether and how environmental barriers influence the capacity and performance of patients with schizophrenia have not been well studied. Thus, we had two hypotheses. First, the effects of the environment as a whole on the daily activity performance in patients with moderate and severe schizophrenia differ because they have different severity in activities of daily living impairment [22]. In addition, second, using a WHODAS 2.0 score, we could determine which patients with schizophrenia would experience an e1 environmental barrier. Therefore, we aimed to provide a comprehensive survey of the functional outcomes of capacity, performance, and the capacity-performance discrepancy as well as the relationship between functional outcomes and e1 environmental barriers in schizophrenia.

## 2. Materials and Methods

### 2.1. The Reconstruction of the Health and Social Welfare System in Certification of Disability and Care Needs in Taiwan

Since July 2012 in Taiwan, the People with Disabilities Rights Protection Act has specified that the assessment for disability and care needs be made according to the ICF model [23]. Following the establishment of the MOHW in 2013 in Taiwan, the Department of Health integrated with the social welfare agencies within the Ministry of the Interior to form a new national organization responsible for social welfare named the Social and Family Affairs Administration (SFAA), MOHW, Taiwan. Among the aims of this unification of national organizations was synthesizing disability evaluations with the needs assessment for healthcare and social welfare services to facilitate social participation of people with disabilities. In this study, we used the Data Bank of Certification of Disability and Care Needs from SFAA, MOHW, Taiwan, which collects data on candidates nationwide who want to apply for government benefits or social welfare. The certification of disability and care needs is required to be completed by two or more authorized specialists, including medical doctors and paramedical specialists, such as social workers, occupational therapists, physical therapists, speech therapists, psychologists, and nurses. 

### 2.2. Study Participants and the Data including in the Data Bank of Certification of Disability and Care Needs

We included adults aged ≥ 18 years with schizophrenia (International Classification of Diseases (ICD)-9-CM diagnostic code 295; ICD-10-CM diagnostic codes F20, F25.0–1, F25.8–9, F20.81, and F20.89) with b code b122, who were registered between 1 January 2015 and 31 December 2019. There are 6 domains encompassing 36 items in the WHODAS 2.0, including cognition (6 items), mobility (5 items), self-care (4 items), getting along (5 items), life activities ((domain 5-1) household activities, 4 items; (domain 5-2) work and school activities, 4 items), and social participation (8 items). According to the WHODAS 2.0 manual [24], the 32-item version without the paid work items is still valid for those patients who are not employed. One of the trained paramedical specialists conducted the WHODAS 2.0 to assess each patient’s level of difficulty in doing the items listed within the six domains during the assessment for disability and care needs. A 5-point Likert scale was used with a score of 1 indicating an item that was accomplished without difficulty to a score of 5 indicating an item that was accomplished with extreme difficulty. The comprehensive access barriers to e1 ‘products and technology’ were evaluated in all patients (access barrier = 1, open access = 0), including environmental categories e110 for personal consumption, e115 for personal usage in daily living activities, e120 for personal outdoor and indoor mobility and transportation, e125 for communication, e130 for education, and e165 for objects of economic exchange [25] (Appendix A).

The exclusion criteria included: (1) missing sociodemographic data; (2) missing WHODAS 2.0 domain 1, domain 2, domain 3, domain 4, domain 5-1, and domain 6 data; and (3) missing access barriers to e1 products and technology data. We further categorized the patients into moderate or severe schizophrenia groups according to their GAF score: 31–60 (b122.1–2, moderate schizophrenia) and 1–30 (b122.3–4, severe schizophrenia) [26] (Figure 1). The sociodemographic data including age, sex, primary caregiver, educational level, institution, level of urbanization [27], working status, and economic status of the patients’ families as well as environmental accessibility barriers in the Data Bank of Disability Certification and Care Needs were collected in this study.

### 2.3. The Measurements of Summary Index (SI) of the WHODAS 2.0 Domain and the Capacity-Performance Discrepancy

Based on the formula illustrated in the WHODAS 2.0 manual, we measured the summary index (SI) of each WHODAS 2.0 domain. We could not compute the SI of domain 5-2 (work and school activities) because only 162 patients in the study were employed. The range of the SI was from 0 (least difficulty) to 100 (most difficulty). We also computed the overall SI using the 32 items without the paid work items. There was no missing value in WHODAS 2.0 domain 1, domain 2, domain 3, domain 4, domain 5-1, domain 6, and access barriers to e1 products and technology. The capacity-performance discrepancy was evaluated by the relative difference (RD) for each domain and the SI of functional outcomes using the formula: (the score of capacity—the score of performance)/(the score of capacity + 1 point) [17]. The SIs of the capacity and performance scores of each category of the environment chapter were compared, and the RD were measured to indicate the degree of the category of environmental barriers limiting the daily activity performance. In this study, we hypothesized that patients with schizophrenia having a larger RD encounter more difficulty in activities of daily living if they are undergoing the impact of environmental barriers.

### 2.4. Statistical Analysis

Differences between patients with schizophrenia with moderate and severe impairments and between environment accessibility with and without barriers were assessed using an independent *t*-test for continuous variables and Pearson’s chi-squared test or Fisher’s exact test for categorical variables. Logistic regression was used to test the explanatory power of the factors and compute the odds ratio (OR) of each factor included in the Data Bank of Certification of Disability and Care Needs. Access barriers with a lower bound of 95% confidence interval (CI) of c-statistic >0.7 were considered adequate fit and acceptable to draw inferences from the dataset [28]. We applied a receiver operating characteristic (ROC) curve to determine the best probabilistic cutoff value for the SI performance score to classify patients experiencing an access barrier to e1 categories. We compared the RD in each category between patients with and without an access barrier to e1 categories, stratified for moderate and severe patients using a Mann–Whitney U test. According to Chang et al. [17], the mean RD values were estimated to be 0.06 and 0.04 in the patients with environmental access barrier group and the open access group, respectively, and the SDs were estimated to be 0.06 and 0.04, respectively. The effect size d was 0.39. For a 95% power at a 5% two-tailed significance level with the Mann–Whitney test (two groups), we required the complete data from at least 356 cases (178 in each study group) in order to detect significant differences in the RD with power analysis using G*Power 3.1 [29]. Data were analyzed using the Statistical Package for the Social Sciences (ver. 9.4, SAS, Institute Inc., Cary, NC, USA). Differences between groups were considered significant if the two-tailed *p* values were <0.05.

## 3. Results

During the study period, there were 61,270 people who applied for disability certification. According to the inclusion criteria, 4439 participants who were diagnosed with schizophrenia with b code b122 were included. In accordance with the following exclusion criteria, 557 patients were excluded due to: (1) missing educational data (*n* = 297); (2) missing caregiver data (*n* = 315); and (3) missing family economic status (*n* = 286). After excluding those 557 patients, there were a total of 3882 patients with schizophrenia left (Figure 1).

The severity of schizophrenia did not differ substantially between the male and female patients. Compared with the severe patients, moderate patients were younger, residing in a community with a primary caregiver, employed, and had a better general family economic status (Table 2). Moreover, severe patients had higher average capacity and performance SI scores than moderate patients in the cognition (domain 1), mobility (domain 2), self-care (domain 3), getting along (domain 4), life activities (domain 5-1 household activities), and social participation (domain 6). Therefore, patients with severe schizophrenia experience more difficulty in all daily activities than patients with moderate schizophrenia. The WHODAS 2.0 SI scores were reliable in this study because the Cronbach’s *α* of the items within the SI were 0.965 (capacity) and 0.962 (performance) (*n* = 3882). The percentages of chapter e1 accessibility with barriers were 29.5% in patients with moderate schizophrenia and 37.8% in patients with severe schizophrenia, and the percentages of chapter e1 accessibility with barriers, including categories e110, e115, e120, e125, e130, and e165 were all significantly higher in patients with severe schizophrenia than in patients with moderate schizophrenia (Table 2).

The summarized results of the logistic regressions of category e1 accessibility are shown in Appendix A. The lower bounds of the 95% CI of categories e110, e115, and e120 in all participants were above 0.7 and indicated sufficient evidence for drawing further inferences from the dataset. For category e110 accessibility in patients with schizophrenia (C statistic = 0.765, 95% CI = 0.739–0.792), the existence of category e110 accessibility barriers in patients with schizophrenia was associated with SI score for performance (adjusted OR = 1.04, 95% CI = 1.04–1.05) and middle low-low family economic status (adjusted OR = 1.34, 95% CI = 1.05–1.7). For category e115 accessibility in patients with schizophrenia (C statistic = 0.761, 95% CI = 0.737–0.785), the existence of category e115 accessibility barriers in patients with schizophrenia was associated with SI score for performance (adjusted OR = 1.04, 95% CI = 1.04–1.05), suburban areas (reference category of rural areas, adjusted OR = 1.33, 95% CI = 1.03–1.71), and urban areas (reference category of rural areas, adjusted OR = 1.41, 95% CI = 1.07–1.84). For category e120 accessibility in patients with schizophrenia (C statistic = 0.722, 95% CI = 0.700–0.743), the existence of category e120 barriers in patients with schizophrenia was associated with SI score for performance (adjusted OR = 1.03, 95% CI = 1.03–1.04), lower educational level (adjusted OR = 1.22, 95% CI = 1.01–1.47), and unemployment status (adjusted OR = 2.27, 95% CI = 1.04–4.95).

Table 3 reveals the sociodemographic allocation of the schizophrenia patients with and without accessibility barriers to chapter e1, including e110, e115, and e120. There were higher percentages of age ≥ 65 years, severe schizophrenia, educational level ≤ primary, residence of institution, living in rural and suburban areas, unemployment status, and middle low-low family economic status in patients with an accessibility barrier in category e110 compared with patients without an accessibility barrier in category e110. There were higher percentages of age ≥ 65 years, schizophrenia, educational level ≤ primary, residence of institution, living in rural and suburban areas, and unemployment status in patients with an accessibility barrier in category e115 than in patients without an accessibility barrier in category e115. There were higher percentages of age ≥ 65 years, severe schizophrenia, educational level ≤ primary, and unemployment status in patients with an accessibility barrier in category e120 than in patients without an accessibility barrier in category e120.

Using a ROC analysis, in category e110 accessibility barrier, the SI scores for performance had an AUC = 0.786 (95% CI = 0.749–0.824) in patients with moderate schizophrenia and an AUC = 0.703 (95% CI = 0.661–0.746) in patients with severe schizophrenia (Figure 2; Appendix A). Moderate patients with an SI score of 40 points and severe patients with a score of 55 points were prone to encounter the category e110 accessibility barrier. In category e115 accessibility barriers, the SI scores for performance had an AUC = 0.774 (95% CI = 0.742–0.807) in patients with moderate schizophrenia and an AUC = 0.676 (95% CI = 0.641–0.712) in patients with severe schizophrenia (Appendix A). Moderate patients with an SI score of 34 points and severe patients with a score of 55 points were prone to encounter the category e115 accessibility barrier. In category e120 accessibility barriers, the SI scores for performance had an AUC = 0.733 (95% CI = 0.705–0.762) in patients with moderate schizophrenia and an AUC = 0.673 (95% CI = 0.640–0.706) in patients with severe schizophrenia (Appendix A). Moderate patients with an SI score of 34 points and severe patients with a score of 55 points were prone to encounter the category e120 accessibility barrier. According to the Regulations for the Identification of People with Disability, only stable patients entering the chronic stage of schizophrenia, i.e., at least six months after schizophrenia was confirmed, were appropriate to evaluate disability certification. Hence, the data collection was after the index date of schizophrenia diagnosis. The reference group in ROC analysis was the group without the category e110, e115, or e120 barrier. The comparison of the SI of performance in patients with moderate and severe schizophrenia without e110, e115, or e120 accessibility barrier and those without any e110, e115, and e120 accessibility barriers revealed similar SI, respectively (Appendix A).

The RDs were substantially higher in patients with moderate schizophrenia with an accessibility barrier in categories e110, e115, and e120 than in patients with moderate schizophrenia without an accessibility barrier in categories e110, e115, and e120, respectively. On the other hand, the RD was substantially lower in patients with severe schizophrenia with an accessibility barrier in category e120 than in patients with severe schizophrenia without an accessibility barrier in category e120 (Table 4).

The multiple imputation using WHODAS SI of performance to handle missing data in this study, including educational data, caregiver data, and family economic status, revealed the similar results (Appendix A) to those in Table 2. Appendix A shows the sociodemographic allocation of the schizophrenia patients with and without accessibility barriers to chapter e1, including e110, e115, and e120 with multiple imputation, and the results are similar to those in Table 3. The RDs were also substantially higher in patients with moderate schizophrenia with an accessibility barrier in categories e110, e115, and e120 than in patients with moderate schizophrenia without an accessibility barrier in categories e110, e115, and e120, respectively, with multiple imputation (Appendix A).

The results of ROC analyses with multiple imputation (Appendix A) revealed similar results to those in Appendix A. The results of ROC analyses after stratification by sex and age revealed similar AUC in male and female patients 18–64 years (Appendix A) as those shown in Figure 2 and Appendix A. The RDs were also substantially higher in patients with moderate schizophrenia in both male and female patients aged 18–64 years with an accessibility barrier in categories e110 and e115 than in patients with moderate schizophrenia without an accessibility barrier in categories e110 and e115, respectively, and in male patients with moderate schizophrenia aged 18–64 years with an accessibility barrier in categories e120 than in patients with moderate schizophrenia without an accessibility barrier in categories e120 (Appendix A).

## 4. Discussion

Utilizing the Data Bank of Certification of Disability and Care Needs from SFAA, MOHW, Taiwan, R.O.C., this study analyzed the SI scores between schizophrenia patients with and without an accessibility barrier to environmental categories in chapter e1 (products and technology). This study also utilized the measurements of the capacity-performance discrepancy (RDs) to evaluate the influences of the chapter e1 environmental barriers on the SI scores and each domain of activities of daily living in patients with schizophrenia. Patients with moderate schizophrenia with categories e110, e115, and e120 accessibility barriers were prone to a higher capacity-performance discrepancy based on SI score (Table 4). However, patients with severe schizophrenia with category e120 accessibility barriers were prone to a lower capacity-performance discrepancy based on SI score (Table 4). Thus, chapter e1 environmental barriers are crucial contextual indicators for patients with moderate schizophrenia that decrease their SI performance score. Furthermore, utilizing ROC methods, we classified those patients with schizophrenia who encountered an accessibility barrier to categories e110, e115, and e120 for each WHODAS 2.0 score, with the percentage accurate classification ranging from 67% to 78% (Figure 2; Appendix A). Thus, the relationship between functional outcomes and environmental barriers is noticeable in patients with schizophrenia.

Our finding that the patients’ average SI score for capacity was higher than their average SI score for performance (Table 2) may indicate the existence of environmental barriers for patients with schizophrenia in Taiwan. The chapter e1 environment acts as a barrier for more than 30% of patients with schizophrenia. Notably, in a study from the United States, low-income adults were found to have higher risks of food insecurity that predisposed them to acute medical care [30]. In our study, middle low and low family economic status also increased the risk of barrier in e110, accessibility of products for personal consumption (adjusted OR = 1.34, 95% CI = 1.05–1.7)). Importantly, it has been reported that residents of rural areas are self-reliant and more dependent on family and friends than on healthcare professionals [31]. However, in China, the possibility of being disabled with schizophrenia is somewhat lower among residents of rural areas than among residents of urban areas (adjusted OR = 0.92, 95% CI = 0.86–0.98) [32]. In our study, living in urban and suburban areas also increased the risk of barrier in category e115, accessibility of personal usage in activities of daily living (adjusted OR = 1.41, 95% CI = 1.07–1.84, and 1.33, 95% CI = 1.03–1.71, respectively). In Uganda, low levels of formal education and unemployment increased the risk of barriers to care resulting from transportation difficulties [33]. In our study, lower educational level and unemployment status also increased the risk of barriers in category e120 accessibility of personal outdoor and indoor mobility and transportation (adjusted OR = 1.22, 95% CI = 1.01–1.47, and 2.27, 95% CI = 1.04–4.95, respectively). Our results are also compatible with the disclosure of the United Nations Convention on the Rights of Persons with Disabilities, which found that people with disabilities who have low educational levels and low economic status require more environmental assistance than those with high educational levels and high economic status [34]. However, the causal relationships between the sociodemographic characteristics mentioned above and environmental barriers remain uncertain in this cross-sectional study. Future longitudinal studies are needed to investigate causal relationships and clarify the relationships between other chapters of environmental barriers and facilitators, such as support, attitudes, and services (ICF chapters e3, e4, and e5), and the functional outcomes for patients with schizophrenia.

We also found that the capacity-performance discrepancy was significantly higher in patients with moderate schizophrenia with categories e110, e115, and e120 accessibility barriers than in patients with moderate schizophrenia without categories e110, e115, and e120 accessibility barriers. Multiple environmental barriers may also induce psychological and oxidative stress [35] and be associated with functional decline and senescence [36]. While this finding is similar to those of previous studies assessing other chronic disabling conditions [17], we found that the capacity-performance discrepancy was significantly lower in patients with severe schizophrenia with category e120 accessibility barriers than in patients with severe schizophrenia without the category e120 accessibility barrier. Notably, patients with severe schizophrenia without primary caregivers were more likely to be in residential care facilities. Upon further analysis in severe schizophrenia and category e120 with accessibility barriers, we found that institutional residency substantially lowered the capacity-performance discrepancy (Appendix A). Access to transportation is essential for patients with schizophrenia, allowing them to shop, attend health care appointments, and participate in recreational activities. However, when a person’s ambulation function deteriorates, this basic activity becomes an important obstacle. Previous studies have revealed that having access to local recreation and shopping helps mental illness patients feel comfortable residing in residential care facilities [37]. However, in residential care facilities, healthcare workers continue to assist severe patients during the process of activities of daily living. This assistance may decrease the capacity-performance discrepancy and improve mental and physical health [38]. Nevertheless, this cross-sectional study did not determine a causal relationship between the capacity-performance discrepancy and environmental barriers. Future studies are needed to follow-up longitudinally for the ascertainment of whether these relationships are causal.

Overall, the major strength of this study is that it contributes a comprehensive assessment of environmental barriers based on the WHODAS 2.0 to evaluate the chapter e1 products and technology. We found that the presence of chapter e1 accessibility barriers was related to functional outcomes. According to the results, policymakers and healthcare professionals should take notice of e1 barrier-free environments to decrease the requirements of schizophrenic patients with functional impairments in the community and in residential care facilities.

### Study Limitations

We analyzed patients with schizophrenia from a nationwide, population-based data bank using a well-known WHO assessment instrument to predict the risk of environmental barriers. However, some limitations of this study must be elucidated. First, the interviewees’ responses to the WHODAS 2.0 instrument may be biased. Because of the cognitive impairment encountered by some patients, caregivers represented some of the patients for the instrument interviews. Therefore, the difference in the respondents (i.e., the instrument was completed either by the patient or the caregiver) might have biased the outcomes. Second, the onset of schizophrenia, treatment history, and presently used therapy in each patient was not noted in the Data Bank of Certification of Disability and Care Needs, only the date of evaluation. Based on the Regulations for the Identification of People with Disability, however, only stable patients entering the chronic stage of schizophrenia can be used to evaluate disability evaluation and certification in Taiwan. Nevertheless, the patients were evaluated at least six months after schizophrenia was confirmed. Third, this study used a cross-sectional design. No further follow-up of the WHODAS 2.0 instrument responses from the patients was conducted; thus, future studies are required to address this issue. We use multiple imputation to handle the missing data and stratification by sex and age to control potential confounding factors in the baseline characteristics and found similar results in comparison with the original results. However, it is impossible to draw a clear conclusion on environmental barriers and functional outcomes in patients with schizophrenia in this study because the cross-sectional study design and still other potential confounding factors in the baseline characteristics may interfere with the inference of results. Finally, regarding the study resources, the data bank used was limited to Taiwan; racial, cultural, and medical differences worldwide could have a variable influence on the environmental barriers of patients with schizophrenia. However, this is a nationwide study, and the results are not influenced by different regions in Taiwan. Hence the results are still valuable to regions with infrastructure similar to that of Taiwan.

## 5. Conclusions

In conclusion, we provide a comprehensive evaluation of functional outcomes and environmental barriers in patients with schizophrenia. We found that the presence of chapter e1 accessibility barriers was related to functional outcomes. We also found that the capacity-performance discrepancy was higher in moderate schizophrenic patients with accessibility barriers in the categories of chapter e1 than in patients without accessibility barriers. However, the accessibility barrier in category e120 decreased the discrepancy between patients with and without severe schizophrenia, and institutional care was found to be the potentially decreasing factor. In conclusion, providing a more e1 barrier-free environment in communities and residential care facilities is necessary for patients with schizophrenia in order to decrease their disability.

## Figures and Tables

**Figure 1 ijerph-19-00315-f001:**
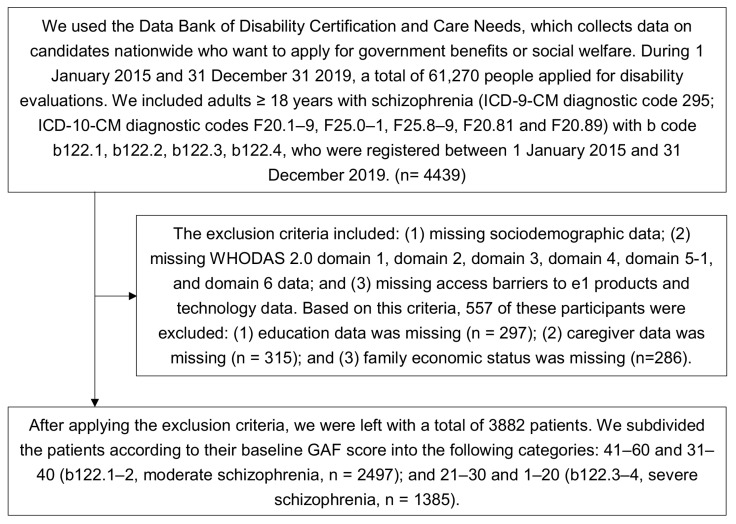
Flowchart of patient selection process. ICD: International Classification of Diseases; WHODAS: World Health Organization Disability Assessment Schedule; GAF: Global Assessment of Functioning.

**Figure 2 ijerph-19-00315-f002:**
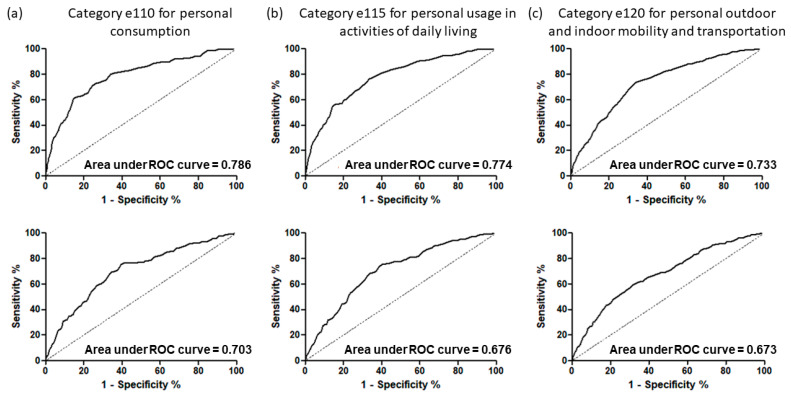
ROC curves used to classify patients with schizophrenia with and without accessibility barriers in the categories of chapter e1 products and technology by utilizing the summary index scores of their performance. Results for patients with moderate schizophrenia (*n* = 2497) are presented in the top row, while results for patients with severe schizophrenia (*n* = 1385) are found in the bottom row. Column (**a**) shows results for category e110 for personal consumption, column (**b**) shows results for category e115 for personal usage in activities of daily living, and column (**c**) shows results for category e120 for personal outdoor and indoor mobility and transportation.

**Table 1 ijerph-19-00315-t001:** PECO (Population, Exposure, Comparator, and Outcome) statement in this study.

Element	Evidence
Population	Schizophrenic patients
Exposure	Environmental barriers
Comparison	No environmental barriers
Outcome	More capacity-performance discrepancy

**Table 2 ijerph-19-00315-t002:** Demographics of the study population and WHODAS 2.0 evaluation results.

	Moderate Schizophrenia (*n* = 2497)	Severe Schizophrenia (*n* = 1385)	*p*	Statistics (Statistical Tests)	Degrees of Freedom (df)
Female (*n*, %)	1179 (47.2)	637 (46.0)	0.464	0.536 (chi-squared test)	1
Age (years old, mean (SD))	48.3 (14.4)	55.9 (13.5)	<0.001	16.490 (*t*-test)	3001.4
Education			<0.001	72.658 (chi-squared test)	1
>Primary	1678 (67.2)	739 (53.4)			
≤Primary	819 (32.8)	646 (46.6)			
Residence			<0.001	321.747 (chi-squared test)	1
Community	1066 (42.7)	201 (14.5)			
Institution	1431 (57.3)	1184 (85.5)			
Primary caregiver			<0.001	94.701 (chi-squared test)	1
Yes	801 (32.1)	244 (17.6)			
No	1696 (67.9)	1141 (82.4)			
Urbanization level			0.037	6.621 (chi-squared test)	2
Rural	560 (22.4)	361 (26.1)			
Suburban	707 (28.3)	380 (27.4)			
Urban	1230 (49.3)	644 (46.5)			
Work status			<0.001	75.311 (chi-squared test)	1
Employment	156 (6.3)	6 (0.4)			
Unemployment	2341 (93.8)	1379 (99.6)			
Family economic status			<0.001	54.731 (chi-squared test)	1
General	1464 (58.6)	641 (46.3)			
Middle low-low	1033 (41.4)	744 (53.7)			
WHODAS 2.0 (mean (SD))					
Cognition (domain 1)					
Capacity	38.2 (25.3)	60.6 (28.4)	<0.001	24.449 (*t*-test)	2515.8
Performance	36.1 (24.4)	58.1 (28.4)	<0.001	24.271 (*t*-test)	2591
Mobility (domain 2)					
Capacity	20.4 (28.5)	40.4 (37.0)	<0.001	17.504 (*t*-test)	2254.8
Performance	17.7 (25.0)	35.3 (33.5)	<0.001	17.102 (*t*-test)	2305.2
Self-care (domain 3)					
Capacity	18.9 (25.6)	41.9 (34.6)	<0.001	21.738 (*t*-test)	2096
Performance	14.5 (20.8)	31.7 (30.7)	<0.001	18.616 (*t*-test)	2236.5
Getting along (domain 4)					
Capacity	39.8 (25.5)	56.2 (28.8)	<0.001	17.633 (*t*-test)	2551.1
Performance	39.0 (25.2)	54.8 (28.8)	<0.001	17.127 (*t*-test)	2577.5
Life activities (domain 5-1)					
Capacity	44.3 (32.8)	66.3 (36.9)	<0.001	18.457 (*t*-test)	2486.2
Performance	40.8 (32.4)	61.3 (38.2)	<0.001	16.741 (*t*-test)	2588
Social participation (domain 6)					
Capacity	36.4 (23.2)	48.8 (26.7)	<0.001	14.526 (*t*-test)	2500.1
Performance	34.4 (22.1)	45.5 (25.9)	<0.001	13.422 (*t*-test)	2536.8
Overall summary index (SI)					
Capacity	33.4 (21.6)	52.0 (25.4)	<0.001	23.101 (*t*-test)	2427.7
Performance	31.0 (19.9)	47.9 (24.2)	<0.001	22.151 (*t*-test)	2495.7
Chapter e1	737 (29.5)	523 (37.8)	<0.001	27.634 (chi-squared test)	1
Category e110	174 (7.0)	170 (12.3)	<0.001	31.056 (chi-squared test)	1
Category e115	215 (8.6)	203 (14.7)	<0.001	33.901 (chi-squared test)	1
Category e120	319 (12.8)	285 (20.6)	<0.001	41.278 (chi-squared test)	1
Category e125	163 (6.5)	177 (12.8)	<0.001	43.574 (chi-squared test)	1
Category e130	227 (9.1)	177 (12.8)	<0.001	13.002 (chi-squared test)	1
Category e165	537 (21.5)	391 (28.2)	<0.001	21.150 (chi-squared test)	1

**Table 3 ijerph-19-00315-t003:** The sociodemographic allocation and comparison of the patients with schizophrenia without and with accessibility barriers in the categories of chapter e1 products and technology.

Parameters	e110 without Accessibility Barrier	e110 with Accessibility Barrier	*p*	e115 without Accessibility Barrier	e115 with Accessibility Barrier	*p*	e120 without Accessibility Barrier	e120 with Accessibility Barrier	*p*
Total	3538 (91.1)	344 (8.9)		3464 (89.2)	418 (10.8)		3278 (84.4)	604 (15.6)	
Age groups			<0.001			0.005			<0.001
18–64 years	2935 (83.0)	259 (75.3)		2871 (82.9)	323 (77.3)		2729 (83.3)	465 (77.0)	
≥65 years	603 (17.0)	85 (24.7)		593 (17.1)	95 (22.7)		549 (16.8)	139 (23.0)	
Impairment			<0.001			<0.001			<0.001
Moderate	2323 (65.7)	174 (50.6)		2282 (65.9)	215 (51.4)		2178 (66.4)	319 (52.8)	
Severe	1215 (34.3)	170 (49.4)		1182 (34.1)	203 (48.6)		1100 (33.6)	285 (47.2)	
Sex			0.069			0.380			0.122
Male	1899 (53.7)	167 (48.5)		1852 (53.5)	214 (51.2)		1762 (53.8)	304 (50.3)	
Female	1639 (46.3)	177 (51.5)		1612 (46.5)	204 (48.8)		1516 (46.3)	300 (49.7)	
Primary caregiver			0.063			0.681			0.460
Yes	967 (27.3)	78 (22.7)		936 (27.0)	109 (26.4)		875 (26.7)	170 (28.2)	
No	2571 (72.7)	266 (77.3)		2528 (73.0)	309 (73.9)		2403 (73.3)	434 (71.9)	
Education			<0.001			<0.001			<0.001
>Primary	2234 (63.1)	183 (53.2)		2191 (63.3)	226 (54.1)		2087 (63.7)	330 (54.6)	
≤Primary	1304 (36.9)	161 (46.8)		1273 (36.8)	192 (45.9)		1191 (36.3)	274 (45.4)	
Residence			0.002			0.042			0.106
Community	1180 (33.4)	87 (25.3)		1149 (33.2)	118 (28.2)		1087 (33.2)	180 (29.8)	
Institution	2358 (66.7)	257 (74.7)		2315 (66.8)	300 (71.8)		2191 (66.8)	424 (70.2)	
Urbanization level			0.002			<0.001			0.061
Rural	830 (23.5)	91 (26.5)		804 (23.2)	117 (28.0)		764 (23.3)	157 (26.0)	
Suburban	970 (27.4)	117 (34.0)		950 (27.4)	137 (32.8)		905 (27.6)	182 (30.1)	
Urban	1738 (49.1)	136 (39.5)		1710 (49.4)	164 (39.2)		1609 (49.1)	265 (43.9)	
Work status			0.003			0.003			<0.001
Employed	158 (4.5)	4 (1.2)		156 (4.5)	6 (1.4)		155 (4.7)	7 (1.2)	
Unemployed	3380 (95.5)	340 (98.8)		3308 (95.5)	412 (98.6)		3123 (95.3)	597 (98.8)	
Family economic status			0.002			0.128			0.624
General	1946 (55.0)	159 (46.2)		1893 (54.7)	212 (50.7)		1783 (54.4)	322 (53.3)	
Middle low-Low	1592 (45.0)	185 (53.8)		1571 (45.4)	206 (49.3)		1495 (45.6)	282 (46.7)	

**Table 4 ijerph-19-00315-t004:** Comparison of relative difference (RD, capacity-performance discrepancy) of the summary index (SI) between schizophrenia patients with and without accessibility barriers to the categories in chapter e1 products and technology, including environmental categories e110 for personal consumption, e115 for personal usage in activities of daily living, and e120 for personal outdoor and indoor mobility and transportation, stratified for patients with moderate schizophrenia (*n* = 174 in accessibility of e110 with barrier, *n* = 215 in accessibility of e115 with barrier, *n* = 319 in accessibility of e120 with barrier) and severe schizophrenia (*n* = 170 in accessibility of e110 with barrier, *n* = 203 in accessibility of e115 with barrier, *n* = 285 in accessibility of e120 with barrier).

	Moderate Schizophrenia (*n* = 2497)	Severe Schizophrenia (*n* = 1385)
Accessibility of e110		
with barrier	0 ** (0–9.34)	0.57 (0–9.65)
without barrier	0 ** (0–4.36)	0 (0–10.31)
Accessibility of e115		
with barrier	0 ** (0–9.42)	1.09 (0–9.60)
without barrier	0 ** (0–4.45)	0 (0–10.31)
Accessibility of e120		
with barrier	0 ** (0–11.56)	0 * (0–9.31)
without barrier	0 ** (0–3.39)	0 * (0–10.53)

Relative difference = (SI score of capacity – SI score of performance)/(SI score of capacity + 1 point). Median (interquartile range (IQR)). Mann–Whitney U test; * *p* value < 0.05, ** *p* value < 0.01.

## Data Availability

Data are available from the Social and Family Affairs Administration, Ministry of Health and Welfare, Taiwan, for researchers who meet the criteria for access to confidential data for academic purposes with a permission letter. The IRB of NCKUH is entitled and has full rights to oversee all activities of each researcher to comply with the Personal Data Protection Act. Anyone interested in analyzing the same dataset must write a research proposal with full protection of human rights and apply to the IRB of NCKUH to obtain access.

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
