# Peer review of "Environmental Barriers and Functional Outcomes in Patients with Schizophrenia in Taiwan: The Capacity-Performance Discrepancy"

_ijerph, 2021, doi:10.3390/ijerph19010315_

Round 1

Reviewer 1 Report

The paper “Environmental barriers and functional outcomes in patients with schizophrenia in Taiwan: the capacity -performance discrepancy”, written by Lien et al., considers environmental barriers in schizophrenic patients’ life. The article is interesting well written. However, there are some concerns related to the chosen patients. There is not clear if treatment history and presently used therapy were also considered in the investigation and analysis.

Reviewer 2 Report

This study aims to investigate whether functional outcome and capacity-performance discrepancy in patients with schizophrenia are related to environmental barriers, specifically the environmental chapter 1 (e1): products and technology. The topic can be of clinical importance in that the chronic schizophrenia patients are prone to functional disability in a wide range of domains, and the environmental barriers may affect the functional outcome and related social burden. The study design is sound and the description in the manuscript is well written with sufficient details. I appreciate the work by the authors. However, I have following concerns that needs to be addressed: 

  1. It would be better to provide the statistic values (e.g.t, F, chi-square,e.t.c.) and degree of freedom on the Tables and on the result manuscript. Providing only the p-value is somewhat insufficient, especially in the case that most compared variables show high significance indicated as the same ‘p<0.001’. 
  2. At the 1st paragraph of the Discussion section, the authors claim that they ‘successfully’ classified the barriers of schizophrenia patients. The term ‘successful’ to indicate the level of classification accuracy is subjective and ambiguous. Furthermore, I will not disagree with theLogRegdesign that the authors are employing, but would not say that AUROC around 0.7 for a binary classification task is somewhat ‘successful’. Please consider removing or changing the adjective to a more objective one. 
  3. The authors summarize their statistical findings at the 2nd, 3rd paragraph of the Discussion section. When summarizing the results, the authors should note a caution to the readers that the causation of environmental barrier -> functional outcome (or other domains) cannot be elucidated from this study results because many factors may have confounded the significance of the statistical tests. That said, there are many (actually all except the gender variable) significant results with p<0.05 from Table 1. which means that the difference between the two groups can be confounded by any significant variable included in the analysis. Please discuss this possibility of confounding effect and non-causality in the discussion section.
  4. Adding subtitles for each e1 domain per column can increase the readability of the Figure 2.
  5. Some sentences are confusing and requires further editing, e.g.:
    - the People with Disabilities Rights Protection Act has required that the assessment for disability and care needs be made according to the ICF model (required that the assessment be made?)
    - Completing the certification of disability and care needs requires two or more ~~ (needs requires?)
    - The comprehensive access barriers to e1 products and technology were evaluated (distinguishing ‘products and technology’ within the sentence would be less confusing) 

Reviewer 3 Report

Thank you for the opportunities to review the manuscript.

I think that it is necessary to revise the manuscript. In particular, the study design is not quite good.

1) Is this a cross-sectional study, or a diagnostic study? Please explain this study by using the PECO format.

2) Please fill in the STROBE checklist and/or STARD checklist.

3) Why were participants with missing data excluded? This might cause selection bias. It is necessary to show how to handle missing data in the statistical analysis section including last observation carried forward, multiple imputation and so on.

4) What is confounders, predictors, and effect modifiers in this study? Please explain it to me how to handle these factors in statistical analysis? There are many statistical significant difference in baseline characteristics. 

5) Please explain how sample size was calculated?

6) If you describe the ROC curves, please explain whether data collection was planned before or after the index test and reference standard were performed.

7) In the introduction section (not in the method section), please show both the aim and hypothesis of this study.

8) what is the definition of Environmental barriers in this study? The authors hypothesized in the method section that patients with schizophrenia having a larger RD encounter more difficulty in activities of daily living if they are 
undergoing the impact of environmental barriers. Why will the environmental barriers make an influence on the difficulty in activities of daily living? Are there any previous studies on the association between environmental barriers and a larger RD? Are there any previous articles on the association between large RD and activity in daily living in patients with schizophrenia?

I think that it is impossible to make a clear conclusion on Environmental barriers and functional outcomes in patients with schizophrenia, because how to control confounders was not described in spite of many potential confounding factors in the baseline characteristics.

Round 2

Reviewer 3 Report

Thank you for revising the manuscript.

I think the authors clearly answered my questions.

Please show page numbers (do not only answer yes/no) in each items of the STROBE statement (supplementary file) before publication.
